# Consent for Research on Violence against Children: Dilemmas and Contradictions

**Paula Cristina Martins** * 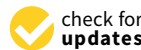 **and Ana Isabel Sani**

Research Centre on Child Studies, University of Minho, 4710-057 Braga, Portugal; anasani@ufp.edu.pt
* Correspondence: pcmartins@psi.uminho.pt

**Abstract:** The increasing visibility of violence involving children has led to a recognition of the need to research its underlying dynamics. As a result, we now have a better understanding of the complexities involved in this kind of research, associated with children's developmental characteristics and social status, exposure to violence, and compromised parenting of caregivers. This paper discusses the issues raised by parental consent in research on violence against children, specifically the dilemma of children's rights to participation and protection, and proposes changes in research practice in this domain.

**Keywords:** child protection; child participation; children's competence to consent; parental consent; research on violence against children; research ethics; victimisation studies

## 1. The Complexities of Research on Violence against Children

The increasing visibility of violence involving children has led to a recognition of the need to research its underlying dynamics. In 2006, the United Nations Study on Violence against Children (VAC) recommended that states should "develop a national research agenda on VAC across settings where violence occurs, including through interview studies with children and parents, with particular attention to vulnerable groups of girls and boys." [1] (p. 29).

Tackling this challenge has led to a better understanding of the complexities involved in this kind of research, associated with:

(a) Children's social status, their perceived vulnerability and incompetence that, on the one hand, casts doubt on the validity of their accounts and decisions and, on the other, may lead to protective measures that prevent their participation in research [2,3];

(b) Contexts where VAC occur, that are difficult to access given their domestic setting that typically involve private adult-child relationships;

(c) Guardianship of adults, usually parents, responsible for protecting and representing children, within the power relationships between them, and possibly conflicting interests [3,4];

(d) Methodological and ethical difficulties resulting from children's developmental characteristics [5];

(e) Methodological and ethical challenges resulting from children's experience of trauma, such as refusal to participate, lack of accuracy or under-reporting [3];

(f) Cultural values of children's families and the importance given to personal autonomy, and their impact on decision-making regarding participation in research [6,7].

Furthermore, as scholars point out, the few ethical guidelines produced for this area [8] tend to portray children involved in this kind of investigation as a homogenous group, characterized by their age and as victims of violence. Yet, the complexity and variability of this population requires consideration:

- On one hand, research on VAC may be conducted with children who are victims or witnesses or in different roles or conditions that may not always be known in advance. Nevertheless, the common child target populations for research in this domain are deemed to be vulnerable children, within a wide spectrum of children's vulnerability profiles, Children may be considered vulnerable because they belong to families at risk, or because they put themselves at risk. They may also be already involved in child protection interventions that are designed to prevent or to reduce the consequences of child abuse or neglect, or to promote their wellbeing [9].

- On the other hand, the forms of victimization, their intensity, duration and frequency, as well as the age at which they occur, are factors that determine significant differences in their experiences and their impacts. Also important is the meaning that children, who are victims of violence, give to victimization events, as well as to their relationship with the offenders. In this sense, considering the history of child victimization seems to be a fundamental requirement for research in this domain [8].

- Furthermore, when we refer to children, we are referring to a population group aged between 0 and 18 years, with a diverse range of needs, forms of expression and relationships [8]. Research procedures and ethical considerations should reflect these differences, not only from the point of view of language comprehension and forms of involvement appropriate to children's interests, but also taking into account that the impact of violence and the impact of participating in research are likely to vary according to the maturity of the child.

- Lastly, the social and cultural characteristics of children's life contexts add further factors of diversity and complexity [8].

Research on VAC combines three characteristics that make it particularly delicate: it is focused on a *sensitive topic* and it involves *children* who may have been *victims*. Despite all precautions, participants in research on VAC are invited to take part in activities that may be "felt as intrusive, uncomfortable to disclose, and socially undesirable "[10] (p. 2) and, as such, be potentially disruptive, distressing or ultimately harmful [8], particularly for the child victims. In addition, as argued elsewhere [2], the social attributes ascribed to both children and victims are virtually identical, in that both are viewed as vulnerable, powerless and in need of protection. In this sense, the victim's perceived vulnerability reinforces the child's perceived vulnerability. Because of this "convergent negativity (children and victims), social impact of child victimisation is quite significant" [2] (p. 54). The child and the victim are "both voiceless because others speak of them, for them, but usually not to them or because nobody speaks at all about it (violence)" [2] (p. 54).

Therefore, in research on VAC, it must be recognized that to hear from children is not only a research requirement but also an ethical imperative. Either as victims or as witnesses, children are key informants of their lives [11] and experiences; gathering the facts reported by them is as important as understanding their perspectives and the meanings they attach to events. However, it is noteworthy that such research is not neutral or "innocent" [4] (p. 206). It must be conducted with the utmost care, methodological adequacy and ethical integrity, in order "to capture the full account of children's views and perspectives" [4] (p. 206) and not misinterpret their answers and silences. Otherwise, poor quality research practices involving children may paradoxically compromise the value of their participation, eventually resulting "in manipulation, decoration or tokenism." [7] (p. 4) and thus continuing to keep children out of reach [4] in that they are alienated both in and from the research process. These features combine to make research with children on sensitive topics, particularly violence, fraught with challenges [10].

## 2. Parental Permission for Children's Participation in Research

Parental consent is at the core of the ethical and methodological debate concerning children's participation in research. Giving parental permission is considered simultaneously as a right, a duty, a power and a responsibility of parents.

Adults and, especially, parents are responsible for defending the best interests of their children. Because they usually have a unique relationship with their children, formed by an affective bond, that gives them privileged knowledge, they are key actors in defending their child's well-being [5]. However, as Hagger argues, the assumption that parents are always in the best position to make the most appropriate decisions on behalf of their children is not always true [5]. In fact, parents may not have enough information to make decisions, or may decide according to their own interests and views, disregarding their children's perspective. Furthermore, as Hagger points out, research is a complex process with many implicit aspects that are unpredictable and, as such, may not be anticipated by parents [5]. Ultimately, parents' interests may be in conflict with those of their children.

Parents may not authorize the participation of their children in research on sensitive topics, such as child maltreatment or family violence [12], for a variety of reasons [8]:

- to preserve their family's privacy and thus prevent the child from revealing unintended information, in so far as parents are indirect subjects in VAC research [13];
- to defend what they consider to be their children's best interests, namely to shield them from: (i) experiencing discomfort or being exposed to distressing situations (e.g., experiences of boredom, inconvenience, stress, fear of failure, lowering of self-esteem, guilt, embarrassment) [13]; (ii) suffering potential harmful consequences engendered by the research process such as re-traumatisation or the risk of confidentiality being breached; (iii) being negatively labelled due to their involvement in research (the project, its aim and also the recruitment procedure) [9], even if initially they are not formally identified as victims [4];
- to protect their own interests i.e., to conceal their inadequate or harmful behaviour towards their children and thus avoid the consequences.

As a result of adults' self-interest or their interpretation of children's interests, children are subject to relationships of power and control that are expressed in terms of obligations, expectations and prohibitions that may obstruct their participation in research. Sometimes silencing their voice as victims and at other times denying their testimony as witnesses, in any event, these relationships disregard their perspective. Beyond the ethical issues raised, this may lead to a sample bias that could compromise the validity of research results and the development of knowledge about phenomena of social interest [12]. Underlying these concerns is the dialectic between children's rights to protection and participation and the associated concepts of child autonomy and competence [6,14,15]. As stated by Ruiz-Casares et al., "Whereas sometimes the lack of adult involvement can hinder children's and young people's development and access to resources, overprotection of children and young people can result in their exclusion from processes that affect them at the expense of the children and young people themselves and substantial loss for the communities where they live." [7] (p. 4) This raises the problem of the need for parental consent and its possible limits, especially considering two variables: the age of the child and the legitimacy of the parent.

### 2.1. The Problem of Child Incompetence

Although varying across countries [16], typically national laws and regulations are based on the legal age of consent, grounded on the concepts of children's immaturity and incompetence. On the other hand, parents are deemed natural and responsible decision makers for their children. Such static and abstract conceptions of children's and adults' capacities are conventional but arbitrary. In fact, both the legitimacy and power of parents to give their children permission to participate in research and children's heteronomy and incompetence are not considered absolute or universal. On the contrary, as Cashmore contends, there is a remarkable range of opinions [12], either based on developmental and neurological evidence [17] or ethical and methodological arguments [18].

What is at stake is that competence is not age-related [15] nor is autonomy. Childhood is a developmental period that involves continuous maturation, learning, and change of behaviours and capabilities. Children's participation in research should reflect this evolution and their social

involvement [7]. Conversely, children's competence also depends on their experience of participation supported by adults (scaffolding) [15]. Autonomy is also relational in nature; Sabatello et al. refer to this concept as *autonomy with others* to designate a "dynamic process of negotiation" between children and parents [6] (p. 2). According to these views, parents are not proxies for children until they reach the age at which they are legally competent and morally autonomous, but they support the development of their children's competencies, enabling their decision-making. Therefore, similar to what Olszewski and Goldkind argue for medical treatments [19], participation of children in research should be the "default position", and each case should be assessed *per se* [6]. Beyond ethical arguments that could support this position, empirical research reveals that, given appropriate information and time, children's decision-making is comparable to adults [17].

### 2.2. The Limits of Parental Consent

Vulnerable children often find themselves in complex circumstances (e.g., unaccompanied, looked after, runaway or otherwise separated from parents) [8,9] and in situations where families put them at risk and compromise their well-being, as in the case of violence. When parents do not protect their children from harm and, additionally, are a danger to them, these problems are compounded: on the one hand, children are vulnerable due to the risk of harm or to the actual harm suffered; on the other hand, they are also vulnerable as they lack adults that are responsible for representing them and acting as informed mediators.

Violence against children is a situation where there is an acknowledged conflict of interests between parents and children, resulting in a lack of parental protection. Therefore, in obtaining consent for children's participation in research, many authors recognize that parental authorisation is not an unconditional requirement, especially when children have been maltreated [20]. Furthermore, asking for parental consent may place children at risk [8,20], if children are in contact with the violent parent(s) [21].

Some scholars recommend considering the interest of children's involvement in research in terms of the benefit for the individual participant *vs* social benefit. In no circumstances, should the interests of children in general be used to justify the possibility of potential harm to the participating child. Moreover, participants' benefit should prevail over social and scientific interests [20]. However, this cost-benefit analysis should be carefully considered. For children who are victims, participation in research does not always have direct and immediate impact e.g., in terms of reducing the violence they are undergoing or its damaging consequences [22]. Yet, the experience of being listened to, and the opportunities to have their experiences validated and assigned meaning, can have positive effects [22,23] and lead to empowerment [21]. Furthermore, the social impact of research cannot be underestimated. The results of studies can be used to inform policies and professional practices to improve prevention and intervention on VAC that go beyond the immediate participants of the studies themselves, to the benefit of other children [22].

The question that arises under these circumstances is who should give consent for children to participate in research. There is a growing consensus that child protection takes precedence over parental rights [3]. Perry [8] claims that violence and abuse are private matters of social interest. However, according to Koocher & Keith-Spiegel [9], the courts normally only interfere with family relationships long after damage from bad parental decisions has been done.

In view of the obstacles that normally arise in relation to children's participation in research on sensitive topics and particularly on VAC, and given the urgent need for research into these issues, research ethics committees (RECs) play a critical role. Research on VAC requires specialized theoretical and methodological, as well as ethical, knowledge in all stages of the research process, particularly: design, ethical review procedures, informed consent, recruitment, assessment, intervention, and dissemination [24]. RECs are responsible for analysing research projects from the perspective of the rights and risks of the participants involved. Therefore, as Cater and Øverlienb [21] state, RECs should

carefully consider any situation where parents refuse consent for their children's right to participate and thus its potential empowering effects.

Furthermore, it should be noted that informed consent is more than just "a consent form or a legal document", it is "a communication and decision process" [25] (p. 5), with specificities relating to the type of research, its objectives, context and participants. Typically social research involves "a two-way exchange of information between researcher and potential participants" [8] (p. 36); however most research with children implies a "triad" that includes necessary interactions with parents or children's legal representatives [10,26]. Yet, in the case of research on VAC, the informed consent process often involves a dynamic of multiple relationships, usually in a hierarchy of gatekeepers. Though based on ethical standards for children protection, RECs appraisal of research on VAC normally adopts a conventional and legalistic stance; issues related to children's right to participate and be involved in what concerns their own protection and well-being [7,27] are very rarely considered. Following Ruiz-Casares et al., the nature and requirements of effective participation for children and young people in the context of child protection are not resolved and are an on-going area of concern [7].

## 3. Conclusions

Violence against children is attracting increasing interest from researchers. In consequence, in recent years, a large number of studies on this topic have emerged in the academic community. This sensitive topic, with vulnerable participants, creates new dilemmas and challenges where the scientific value of research and the interests of the participants must be carefully weighed and balanced. This involves going beyond the traditional formal approach circumscribed by ethical and legal guidelines. As Cater and Øverlien argue, "research ethics must not be reduced to a number of principles to be handled routinely" [21] (p. 76). In a similar vein, Isles [28] characterises and questions the informed consent process as often reduced to the collection of a signature on a consent form to guarantee subject's participation.

The conservative model of parents consenting on behalf of their children, followed by children's assent, needs to give way to a joint participatory model where children are included from the beginning, according to their competence, and guided by their parents in the process of decision-making. Whenever necessary, the researcher may triangulate this interaction. In cases where parental consent is difficult or dangerous to obtain, if parents "privilege their own understanding of situations over the child's welfare and rights" [21] (p. 72), the intentional use of limited disclosure by the researcher should be considered within strict limits: (i) to enforce the children's right to participate in research; (ii) when children's participation in research involves no more than minimal risk, with the prospect of direct or indirect benefit to subjects; iii) where the extent of limited disclosure is clearly defined [29].

Research on VAC requires the adoption of an attitude of responsibility, vigilance and reflexivity throughout the research process [30]. Moreover, it necessitates a new paradigm of communication between RECs, researchers and research participants; closer, continuous and more horizontal communication will allow researchers to better understand children's perspectives and to be sensitive to their needs [21]. This new paradigm of communication, more flexible and fluid, encompasses *ongoing consent* as suggested by Flewitt [31], involving close attention and response by the researcher to children's reactions, and the relationship of *ethical symmetry* proposed by Christensen and Prout [30], which involves giving children and adults the same status as research participants.

Research projects on VAC ought to provide evidence that they are necessary, valid and ethical. Therefore, compliance with ethical principles and guidelines is a key requirement. However, the role of RECs should not be limited to checking compliance with ethical requirements at the inception of research projects. RECs need to become more closely-coupled to the researchers and continue to monitor the development of research projects and their processes e.g., communication between researchers and participants. This would result in an enhanced understanding of the needs and experiences of researchers [32], the participants' characteristics, and the sociocultural context where research is being conducted. We believe that ongoing ethical deliberation, informed by concrete

knowledge of the research as it develops, will allow for the relevant ethical and methodological issues to be addressed, as necessary, at the various stages of the research process.

Assessment and management of risk of harm for research participants is part of high quality research. Considering that vulnerability is context-dependent [27] and individual competence and autonomy are an expression of meaningful relationships of individuals with contexts, people and processes, quality research does not constitute a risk factor for harm to participants deemed vulnerable [33]. Therefore, instead of adopting a paternalistic approach, concerned with making access to children difficult [33], RECs should work with research teams to enable robust and adaptive research programmes. Only the adoption of a dynamic, contextual and personalized approach guarantees that the children's involvement in research is appropriate. This is necessary for accessing and for understanding the experience of the key informants (and beneficiaries) of this domain. By doing so, we give children the opportunity to contribute to the research agenda, to improve our research processes, and, indirectly, to influence socio-political changes based on research evidence [25] that have the potential for positive impacts on child victimisation and interpersonal violence [22].

**Author Contributions:** P.C.M. was involved in the conceptualization and manuscript writing. She is also responsible for the final revision. A.I.S. was involved in the conceptualization of this manuscript. All authors have read and agreed to the published version of the manuscript.

**Funding:** The author(s) disclosed receipt of the following financial support for the research, authorship, and/or publication of this article. This study had financial support from National Funds through the FCT (Foundation for Science and Technology) through CIEC (Research Centre on Child Studies), with the reference UID/CED/00317/2019.

**Conflicts of Interest:** The authors declare no conflict of interest.

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
