# Peer review of "Consent for Research on Violence against Children: Dilemmas and Contradictions"

_societies, doi:10.3390/soc10010015_

Round 1

Reviewer 1 Report

This paper is a commentary rather than research. I believe the topic is very important but your argument lacks clarity and many sentences do not make sense., (for example, Line 11 & 12) "By invoking the dilemma of children's rights to participation and protection, it is argued that child protection is mediated by their participation in social context and not by their exclusion". I think you are trying to say something like: "The rights of children to be actively given a voice in matters which directly concern them serves to enhance rather than detract from their welfare and protection overall" - would this be close??

I would suggest getting help from someone who can help with re-editing the paper. This is a very contentious but important issue and deserves to have more adult voices advocating for the rights of children to provide their perspective, as well as providing some suggestions for just how this might be achieved, and safely, particularly for children who live in families where there is ongoing violence but may seldom be asked what this is like for them.

You can offer a valid contribution, particularly in relation to the limitations of parental consent regarding children's right to participate.

Author Response

The authors would like to thank the Reviewer for these comments. We appreciate the time committed to our paper and your comments and suggestions.

A native English speaker proofread the manuscript. We think that our paper gained in clarity, consistency and accuracy.

Conclusions were extensively reviewed to include more recommendations and practical suggestions, particularly, regarding limitation of parental consent and strategies to go beyond it, in specific circumstances.  

Attached is our revised manuscript with all revisions highlighted to illustrate how and where it has been changed.

Reviewer 2 Report

The authors state that this article is written with the intent of discussing issues related to parental consent in research on violence against children. They introduce the topic by explaining why children’s participation in research is particularly complex, noting that children are often thrust into research in which they are not given a voice or a role in constructing their own narratives. They propose that research on children exposed to violence must be carried out with utmost care and regard for the well-being and safety of those involved.

The authors further explain that parental consent in research on vulnerable children is complicated for a variety of reasons, including that children have little say in deciding whether and how to participate in studies about their own experiences. The dynamics surrounding children’s involvement in research are even more uncertain when parents (and others responsible for their care) consent without primary consideration of their rights and safety. The authors comment that “children’s protection and participation occur in the context of their relationships and in this context should be assessed. Only the adoption of a dynamic, contextual and personalized approach guarantees their appropriate involvement in research” (lines 206-208).

Any number of points raised in this article are important and worthy of debate and discussion, but what follows from the above general recommendation is not immediately evident, nor is it clear what to take from this commentary for further analysis of the referenced complexities. To succeed in writing an article that will have impact on the VAC field, my recommendation is to end with a list of very specific recommendations and series of next steps to advance the conversation.  These recommendations might include guidance on how to determine risks and benefits to children in complex research environments, or possibly, when and in which scenarios more scrutiny by ethics committees regarding children’s safety is required.  Further, in “considering participants' characteristics and the sociocultural context in which research is conducted,” (lines 210-212), what specifically can be done, and how? And, what are ethics committees to do to “enable research teams in order to facilitate research on child victimization” (lines 217-218)?

Author Response

The authors would like to thank the Reviewer for these comments. We appreciate the time committed to our paper and your comments and suggestions.

We understood that parts of our paper lacked clarity and needed to have more concrete orientations, as well as conclusions that make valid contributions to research practice. With this purpose, we did an extensive review of the language in order to better specify and make more concrete our ideas and to advance some recommendations. Conclusions were extensively reviewed.

Attached is our revised manuscript with all revisions highlighted to illustrate how and where it has been changed.

Round 2

Reviewer 2 Report

On the whole, I believe the authors have done a good job of addressing earlier comments and requests for more specificity and greater clarity around key points and recommendations. The argument for transparency and a more inclusive and empowering consent process for children involved in violence research is important and timely.